# Rapid Depletion of Subcutaneous Adipose Tissue during Sorafenib Treatment Predicts Poor Survival in Patients with Hepatocellular Carcinoma

**DOI:** 10.3390/cancers12071795

**Published:** 2020-07-04

**Authors:** Kenji Imai, Koji Takai, Takao Miwa, Daisuke Taguchi, Tatsunori Hanai, Atsushi Suetsugu, Makoto Shiraki, Masahito Shimizu

**Affiliations:** Department of Gastroenterology/Internal Medicine, Gifu University Graduate School of Medicine, 1-1 Yanagido, Gifu 501-1194, Japan; koz@gifu-u.ac.jp (K.T.); takao.miwa0505@gmail.com (T.M.); fiction_may_relieve@yahoo.co.jp (D.T.); hanai0606@yahoo.co.jp (T.H.); asue@gifu-u.ac.jp (A.S.); mshiraki-gif@umin.ac.jp (M.S.); shimim-gif@umin.ac.jp (M.S.)

**Keywords:** body composition, hepatocellular carcinoma, sorafenib, prognostic factor, skeletal muscle, subcutaneous fat mass

## Abstract

The aim of this study was to assess the annualized changes in body composition, including skeletal muscle, subcutaneous adipose tissue (SAT), and visceral adipose tissue (VAT) before, during, and after sorafenib treatment in patients with hepatocellular carcinoma (HCC). This retrospective study evaluated 61 HCC patients treated with sorafenib. Annualized changes (Δ; cm^2^/m^2^/year) in skeletal muscle index (SMI), SAT index (SATI), and VAT index (VATI), which were defined as the cross-sectional areas (cm^2^) of those areas on computed tomography normalized by the square of one’s height (m^2^), before (_pre_), during (_during_), and after (_post_) sorafenib treatment, were calculated. Patients within the 20th percentile cutoffs for these indices were classified into the rapid depletion group and the effects of these values on survival were analyzed using the Kaplan-Meier analysis and Cox proportional-hazards model. Annualized depletion rates of SMI (ΔSMI_pre_: −3.5, ΔSMI_during_: −3.5, ΔSMI_post_: −8.0) and VATI (ΔVATI_pre_: −3.2, ΔVATI_during_: −2.8, ΔVATI_post_: −15.1) accelerated after the cancellation of sorafenib, whereas that of SATI (ΔSATI_pre_: −4.8, ΔSATI_during_; −7.6, ΔSATI_post_; −8.0) had already accelerated during sorafenib treatment. Patients with rapid depletion of ΔSATI_during_ experienced significantly worse survival rates (*p* < 0.001), and it was an independent predictor of survival (*p* = 0.009), together with therapeutic effect (*p* < 0.001). Rapid depletion of SAT during sorafenib treatment can be used to predict survival in patients with HCC.

## 1. Introduction

Hepatocellular carcinoma (HCC) is one of the most common global malignancies; more than half a million people are diagnosed with HCC annually [1]. The prognosis of HCC is notably poor due to difficulties in detecting this malignancy at an early stage [2,3]. Recently, oral molecular targeted drugs have played an important role in the treatment of advanced HCC [4,5]. Sorafenib is the first orally active multi-kinase inhibitor that has been confirmed to be efficacious against advanced HCC [6]. Therefore, sorafenib is now widely used for the treatment of patients with advanced HCC [4,5]; however, several adverse events, including hand–foot syndrome (HFS), fatigue, weight loss, and gastrointestinal events, such as anorexia, diarrhea, nausea, and vomiting often occur when using this agent [6]. Once these adverse gastrointestinal tract events occur, malnutrition can result, causing reductions in body weight and unfavorable changes in body composition, such as sarcopenia, depletion of adipose tissues, and cachexia.

Elevations of energy expenditure and catabolism due to tumor presence and tumor–host interactions are considered to be factors contributing to cancer-associated cachexia in several types of malignancies, including HCC [7]. Patients with advanced cancers frequently show losses in total body weight, skeletal muscle, and adipose tissue [8,9,10]. Therefore, it is easy to imagine that advanced HCC patients treated with sorafenib could be susceptible to decreases in skeletal muscle and adipose tissues, due to both enlarged tumor burden and decreased food intake, caused by digestive-system-related adverse events of the agent.

Several prognostic factors, such as clinical cancer stage, extrahepatic metastasis including bone lesions, and liver functional reserve, are reported in patients with HCC [11,12,13]. Recently, unfavorable changes in body composition, such as sarcopenia and cachexia, have attracted great attention as new prognostic factors for various malignancies, including HCC. For example, skeletal muscle depletion assessed by computed tomography (CT) can predict a poor prognosis in all cancer stages [14] and in sorafenib-treated patients with HCC [15]. Rapid depletions of skeletal muscle mass and subcutaneous adipose tissue (SAT) also predict worse survival rates in patients with HCC treated with sorafenib [16]. Moreover, sarcopenia and rapid skeletal muscle wasting are associated with worse survival rates in patients with liver cirrhosis, who are more susceptible to HCC [17,18].

Because sarcopenia and rapid depletions of skeletal muscle mass and SAT are involved in poor survival rates of HCC patients [14,15,16,17,18], correct assessment of body composition and nutritional therapy based on that assessment are required to improve the prognosis of such patients. However, there has been no study that evaluates changes in body composition in HCC patients treated with sorafenib from the initial stages to death using sequential CT images. The purpose of this study was to compare the annualized changes in body composition, including skeletal muscle, SAT, and visceral adipose tissue (VAT) before, during, and after sorafenib treatment in patients with advanced HCC. Impacts of those annualized changes on survival rates in sorafenib-treated HCC patients were also evaluated.

## 2. Results

### 2.1. Baseline Characteristics and Laboratory Data of HCC Patients Treated with Sorafenib

The baseline characteristics and laboratory data of 61 patients (53 men and 8 women, average age 67.3 years) just before the introduction of sorafenib are shown in Table 1. All patients had good liver functional reserves (Child–Pugh score 5 or 6) and advanced stage HCC (cancer stage more than III), according to the Clinical Practice Guidelines for HCC issued by the Japan Society of Hepatology (JSH) [19]. The therapeutic effect of complete response (CR), partial response (PR), stable disease (SD), and progression disease (PD) was 3, 5, 19, and 34 cases, respectively. Sorafenib was discontinued because of PD (*n* = 22); severe adverse events, including HFS (*n* = 4); general fatigue (*n* = 4), ascites (*n* = 4); liver dysfunction (*n* = 4); appetite loss (*n* = 4); and other reasons. The median duration of sorafenib treatment was 15.7 months (95% confidence interval (CI) 12.3–19.0 months), and the median survival time after sorafenib introduction was 19.8 months (95% CI 15.1–26.1 months) and the 1-, 2-, and 3-year overall survival rates were 68.2%, 38.3%, and 17.2%, respectively. Female patients were significantly younger (*p* = 0.022), had more SATI (*p* < 0.001), and less hepatitis C virus (HCV)-related HCC (*p* = 0.042) when compared to male patients.

### 2.2. Changes in Body Composition before, during, and after Sorafenib Treatment

The values of skeletal muscle index (SMI), subcutaneous adipose tissue index (SATI), and visceral adipose tissue index (VATI) at the time of HCC diagnosis (initial stage), the introduction of sorafenib, the cancellation of sorafenib, and last observation were 46.6, 43.7, 39.9, and 38.9 (cm^2^/m^2^; SMI), 36.8, 34.9, 29.4, and 27.5 (cm^2^/m^2^; SATI), and 37.8, 36.6, 34.0, and 31.2 (cm^2^/m^2^; VATI), respectively. All of these tended to decrease over time, and some of these values showed significant differences (Figure 1).

Mean intervals of pre-sorafenib, duration of sorafenib treatment, and post-sorafenib stages were 2.8, 1.3, and 0.7 years, respectively (Table 2). Annualized changes in ∆SMI, ∆SATI, and ∆VATI of each stage are shown in Figure 2. Annualized depletion rates of SMI (ΔSMI_pre_: −3.5, ΔSMI_during_: −3.5, ΔSMI_post_: −8.0) and VATI (ΔVATI_pre_: −3.2, ΔVATI_during_: −2.8, ΔVATI_post_: −15.1) accelerated at post-sorafenib stage for the first time, whereas that of SATI (ΔSATI_pre_: −4.8, ΔSATI_during_; −7.6, ΔSATI_post_; −8.0) had already accelerated at the sorafenib stage. During the administration of sorafenib (41 cases), and after cancellation of this agent (15 cases), other HCC treatments, such as radiofrequency ablation, transcatheter arterial chemoembolization, and radiation therapy, were conducted on the patients. Sorafenib was temporally stopped when other treatments were introduced, as the safety of combination therapies using sorafenib and any other treatments for HCC is not ensured. The detailed results of the treatments during each stage are shown in Table 2.

### 2.3. Impact of Annualized Changes in Body Composition on Survival in the Patients with HCC Treated with Sorafenib

In order to determine which parameters of annualized changes in body composition can predict prognosis of HCC patients treated with sorafenib, we divided the enrolled patients into two groups based on the 20th percentile cutoffs for each gender, because adipose tissue distribution is known to differ by gender [20]. The patients within the two groups were classified into a rapid depletion (RD) group and a non-rapid depletion (non-RD) group, and compared using the Cox proportional-hazards model. As shown in Table 3, five factors were identified as significant by univariate analysis, including therapeutic effect (PD vs. CR/PR/SD, *p* < 0.001), the presence of sarcopenia (yes vs. no, *p* = 0.032), ΔSMI_during_ (RD vs. non-RD groups, *p* = 0.004), ΔSATI_during_ (RD vs. non-RD groups, *p* = 0.002), and ΔVATI_during_ (RD vs. non-RD groups, *p* = 0.010). Among these factors, therapeutic effect (hazard ratio [HR]: 3.633, 95% confidence interval (CI): 1.811–7.286, *p* < 0.001) and ΔSATI_during_ (HR: 3.366, 95% CI: 1.362–8.321, *p* = 0.009) were independent predictors of survival in HCC patients treated with sorafenib.

The Kaplan-Meier method revealed significantly poorer survival rates according to the log-lank test for patients with a worse therapeutic effect (PD; *p* < 0.001, Figure 3a), and for those in the RD group with ΔSATI_during_ (<−32.51 for women and <−15.49 for men, Figure 3b).

## 3. Discussion

We have previously indicated that skeletal muscle decreases according to worsening liver functional reserve and larger tumor size in HCC [21]. Furthermore, accelerated skeletal muscle and adipose tissue depletions are reported to be associated with poor survival rates in patients with advanced cancers, including HCC [14,15,16,17,18,22,23]. However, evaluation of body composition using CT images was performed only once or twice in these previous studies. Therefore, this is the first study using sequential CT examinations to elucidate how body composition changes from the initial stage to the terminal stage of HCC patients treated with sorafenib, and to evaluate whether these rapid annualized changes lead to poor survival rates. Diagnosing HCC with sequential CT examinations is extremely useful for detailing changes in body composition because it can avoid the effect of weight gain related to edema, ascites, and pleural effusion, which are often seen in patients with advanced HCC.

The results of the present study showed that skeletal muscle mass, SAT, and VAT decreased over time in HCC patients treated with sorafenib. However, depletion patterns of these body compositions were different from each other. After cancellation of sorafenib, most patients could not be treated with any other active treatments for HCC and instead received only palliative care. Higher priority was given to alleviating pain than to providing sufficient nutritional treatment. Because the patients who could not continue sorafenib treatment suffered from both increased tumor burden and decreased food intake, annualized depletion rates of SMI and VATI were the greatest after cancellation of sorafenib. On the other hand, accelerated depletion of SATI had already occurred during sorafenib treatment and occurred earlier than in SMI and VATI. Furthermore, the annualized depletion rates of SATI during sorafenib treatment were an independent predictor, together with therapeutic effect. These findings, together with those of the previous reports [14,15,16,17,18,22,23], may suggest that evaluation of SAT is a useful strategy for the screening of high-risk patients who show poor prognosis in the early phases of HCC therapy.

The definite reason SAT alone rapidly decreased just after introducing sorafenib in advance of skeletal muscle and VAT remains unclear. SAT is known to act as metabolic storage that can accumulate superfluous energy [24]. Therefore, we consider that SAT might first be used as an energy source when a negative energy balance is induced by sorafenib. Sorafenib frequently causes strong adverse effects on the gastrointestinal tract [6]. Indeed, 44%, 21%, and 25% of patients treated with sorafenib experienced anorexia, diarrhea, and general fatigue, respectively, in the present study. If appetite loss or signs of malnutrition associated with sorafenib treatment were observed, appropriate nutrition therapy and the administration of antiemetics should be started as needed. Furthermore, in order to correct the negative energy balance caused by the adverse effects of sorafenib, it may be effective to reduce the dose of sorafenib or discontinue it temporarily without hesitation.

The results of the present study may suggest that rapid depletion of SAT during sorafenib treatment (∆SATI: <−32.51 for women and <−15.49 for men) implies malnutrition associated with poor survival rates in HCC patients. However, in addition to malnutrition, overnutrition must also be avoided, because obesity and related metabolic disorders can promote liver carcinogenesis [25,26,27]. A recent study revealed that patients with higher than 47.2 cm^2^/m^2^ of VATI had a significantly higher risk of recurrence of HCC after curative treatment [28]. Sarcopenia, which was defined as an SMI value ≤38.0 cm^2^/m^2^ for women and ≤42.0 cm^2^/m^2^ for men in Japan [29], is also a significant prognostic factor of HCC. These cutoff values of body composition may be useful in screening HCC patients showing a poor prognosis.

This study has several limitations. First, it was a retrospective, single-center study, and the sample size, particularly the number of female patients, was comparatively small. Second, 67% of the enrolled patients in this study received other HCC therapies in addition to sorafenib and, therefore, the pure effects of sorafenib on changes in body composition were not evaluated. Third, this retrospective study was unable to assess body weight over time. Body weight is the most convenient and non-invasive indicator of nutritional assessment, and loss of body weight is closely associated with cancer-associated cachexia. Therefore, the body composition data obtained by CT imaging should be compared with changes in body weight in future studies. Fourth, muscle radiodensity was not assessed because it cannot be measured with the image analysis software used in this study. Low muscle attenuation was reported to be a prognostic factor for patients with HCC [30]. Moreover, this study did not assess muscle function parameters such as hand grip strength, which is generally considered a diagnostic criterion for sarcopenia [29] and reported to be a prognostic factor for patients with liver cirrhosis [31]. Therefore, it is quite important to evaluate not only the amount of skeletal muscle, but also its quality. The relationship between skeletal muscle quality and changes in body composition over time is also unclear. In order to solve these limitations and to validate the findings of this study, a prospective study involving a larger number of patients enrolled from several centers should be performed.

## 4. Materials and Methods

### 4.1. Patients, Treatment, and Follow-Up Strategy

This retrospective study enrolled 76 patients treated with sorafenib for advanced HCC between May 2009 and December 2017 at Gifu University Hospital. Among the patients, 11 were excluded because they did not take the agent for more than 1 month, and 4 were excluded because they did not undergo sequential CT examination. The remaining 61 patients were analyzed in this study.

The objective of sorafenib introduction was determined according to the Clinical Practice Guidelines for HCC issued by JSH [19]. Each patient’s therapeutic response was judged using dynamic CT, magnetic resonance imaging, or ultrasound every 3 months according to the Response Evaluation Criteria in Cancer of the Liver [32], an appropriate system for the assessment of the post-therapeutic response of HCC to sorafenib [33]. CR, PR, SD, and PD are defined as a 100% tumor-necrotizing effect or a complete reduction in tumor size, a tumor-necrotizing effect or tumor size reduction rate between 50% and <100%, effects other than PR and PD, and tumor growth >25% regardless of the necrotizing effect or emergence of a new lesion, respectively [32]. We decided to discontinue this agent if PD was observed for a certain period of time or severe adverse events occurred. The survival time was defined as the interval from the date of sorafenib introduction to the date of death, or September 2019 for surviving patients. All study participants provided verbal informed consent, which was considered sufficient, as this study followed an observational research design that did not require new human biological specimens. The study design, including this consent procedure, was approved by the ethics committee of the Gifu University School of Medicine (ethical protocol code: 29–26).

### 4.2. Image Analysis of Skeletal Muscle Mass and Subcutaneous and Visceral Fat Mass

Skeletal muscle mass, SAT, and VAT were measured using an enhanced CT image (Discovery CT 750 HD, Revolution CT; GE Healthcare, Milwaukee, WI, USA) that had been taken solely for the purpose of diagnosing HCC. A transverse CT image at the third lumbar vertebra (L3) in the inferior direction was assessed. The muscles in the L3 region were analyzed using SYNAPSE VINCENT software (Fujifilm Medical, Tokyo, Japan), as described previously [34]. The cross-sectional areas of the muscle (cm^2^) at the L3 level computed from each image were normalized by the square of the height (m^2^) to obtain SMI (cm^2^/m^2^). Sarcopenia was defined as an SMI value ≤38.0 cm^2^/m^2^ for women and ≤42.0 cm^2^/m^2^ for men, according to JSH guidelines for sarcopenia [29]. In the same manner, the cross-sectional areas of SAT and VAT (cm^2^) at the umbilical point were measured using a built-in function in the SYNAPSE VINCENT software, which can automatically detect CT level at the umbilical point and also automatically measure the cross-sectional area of SAT and VAT at that CT level. These values were then normalized by the square of the height (m^2^) to obtain SATI (cm^2^/m^2^) and VATI (cm^2^/m^2^), respectively.

The progress of the patients was divided into the following three phases: pre-sorafenib stage, from the time of HCC diagnosis to sorafenib introduction (_pre_); sorafenib stage, from the introduction of sorafenib to the cancellation of the agent (_during_); and post-sorafenib stage, from the cancellation of sorafenib to the last observation (_post_). Annualized changes (Δ; cm^2^/m^2^/year) of SMI, SATI, and VATI in each phase of progression (ΔSMI_pre_, ΔSMI_during_, ΔSMI_post_, ΔSATI_pre_, ΔSATI_during_, ΔSATI_post_, ΔVATI_pre_, ΔVATI_during_, and ΔVATI_post_) were then analyzed. The outline and formula of ΔSMI, ΔSATI, and ΔVATI are shown in Figure 4.

### 4.3. Statistical Analysis

Overall survival time was estimated using the Kaplan-Meier method. Differences between curves were evaluated using the log-rank test. The Cox proportional-hazards model was used to analyze which factors affected overall survival. Statistical significance was defined as *p* < 0.05. All statistical analyses were performed using R ver. 3.3.1. (R Foundation for Statistical Computing, Vienna, Austria; http://www.R-project.org/).

## 5. Conclusions

Skeletal muscle mass, SAT, and VAT decreased over time in HCC patients. Annualized depletion rates of SMI and VATI were greatest after withdrawal of sorafenib, but depletion of SATI had already accelerated during sorafenib treatment. Furthermore, together with the therapeutic effect, rapid depletion of SATI during sorafenib treatment (∆SATI: <−32.51 for women and <−15.49 for men) was an independent predictor of survival.

## Figures and Tables

**Figure 1 cancers-12-01795-f001:**
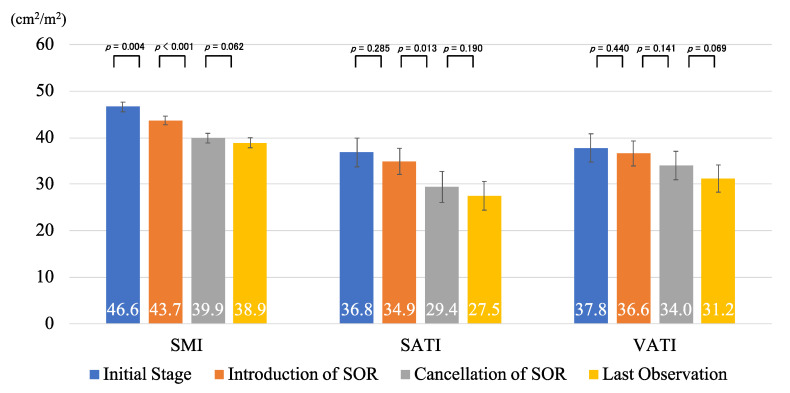
The values of skeletal muscle index (SMI), subcutaneous adipose tissue index (SATI), and visceral adipose tissue index (VATI) at the initial stage, introduction of sorafenib (SOR), cancellation of SOR, and last observation time. Error bars represent standard errors.

**Figure 2 cancers-12-01795-f002:**
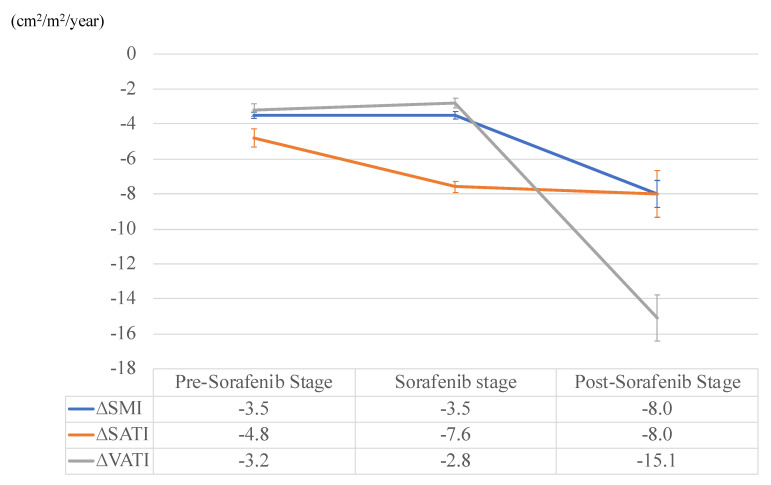
Annualized changes in skeletal muscle index (SMI), subcutaneous adipose tissue index (SATI), and visceral adipose tissue index (VATI) at the pre-sorafenib, duration of sorafenib treatment, and post-sorafenib stages. Error bars represent standard errors.

**Figure 3 cancers-12-01795-f003:**
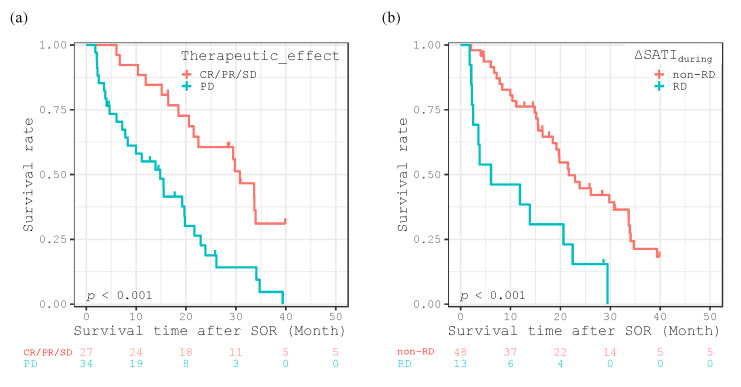
Kaplan-Meier curves for overall survival time after introducing sorafenib (SOR) divided into therapeutic effect (CR/PR/SD vs. PD) (**a**), and the rapid depletion (RD) or non-rapid depletion (non-RD) groups in ΔSATI_during_ (**b**).

**Figure 4 cancers-12-01795-f004:**
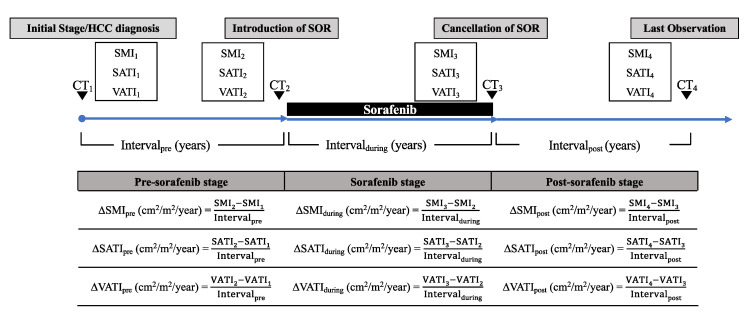
Outline and formula for ΔSMI_pre_, ΔSATI_pre_, ΔVATI_pre_, ΔSMI_during_, ΔSATI_during_, ΔVATI_during_, ΔSMI_post_, ΔSATI_post_, and ΔVATI_post_ (cm^2^/m^2^/year). Skeletal muscle index (SMI) was defined as the cross-sectional area of the muscle (cm^2^) at the L3 level of the computed tomography image, normalized by the square of the height (m^2^). Subcutaneous adipose tissue index (SATI) and visceral adipose tissue index (VATI) were defined as the cross-sectional areas of the subcutaneous and visceral fat (cm^2^), respectively, at the umbilical point, normalized by the square of the height (m^2^).

**Table 1 cancers-12-01795-t001:** Baseline demographic and clinical characteristics of the enrolled patients just before the introduction of sorafenib.

Variables	All (*n* = 61)	Male (*n* = 53)	Female (*n* = 8)	*p*-Value
Age (years)	67.3 ± 11.5	68.9 ± 10.0	59.8 ± 12.4	0.022
Etiology (HBV/HCV/others)	13/28/20	9/27/17	4/1/3	0.042
Child–Pugh score (5/6)	43/18	39/14	3/5	0.094
Cancer stage (III/IVA/IVB)	20/13/28	16/13/24	4/0/4	0.262
BMI (kg/m^2^)	22.3 ± 3.0	22.1 ± 2.7	23.5 ± 4.7	0.218
Extrahepatic metastasis (lung/bone/others)	16/9/9	14/8/8	2/1/1	1.000
SMI (cm^2^/m^2^)	43.7 ± 7.4	44.2 ± 7.3	39.8 ± 6.8	0.118
SATI (cm^2^/m^2^)	34.9 ± 22.0	30.5 ± 17.7	64.2 ± 26.6	<0.001
VATI (cm^2^/m^2^)	36.6 ± 20.9	37.4 ± 21.5	31.8 ± 16.8	0.485
Therapeutic effect (CR/PR/SD/PD)	3/5/19/34	3/3/16/31	0/2/3/3	0.203
Sarcopenia (yes/no)	25/36	22/31	3/5	1.000
Median duration of sorafenib treatment (months)	15.7 (12.3–19.0)	13.3 (12.2–19.5)	9.8 (6.3–22.3)	0.754
Median survival time after introducing sorafenib (months)	19.8 (5.1–26.1)	17.7 (5.5–35.6)	15.8 (7.7–30.3)	0.803

Values are presented as mean ± standard deviation. Sarcopenia was defined as an SMI value of ≤38.0 cm^2^/m^2^ for women and ≤42.0 cm^2^/m^2^ for men. HBV, hepatitis B virus; HCV, hepatitis C virus; BMI, body mass index; SMI, skeletal muscle index; SATI, subcutaneous adipose tissue index; VATI, visceral adipose tissue index; CR, complete response; PR, partial response; SD, stable disease; PD, progression disease.

**Table 2 cancers-12-01795-t002:** Detailed treatment results during the pre-sorafenib, sorafenib, and post-sorafenib stages.

Variables	Pre-Sorafenib Stage	Sorafenib Stage	Post-Sorafenib Stage
Intervals (years)	2.8 ± 2.5	1.3 ± 1.1	0.7 ± 0.5
**Treatment during each stage**			
Surgical resection (none/1/≥2 [times])	28/30/3	61/0/0	61/0/0
RFA (none/1/2/3/≥4 [times])	38/4/11/3/5	58/2/0/1	51/9/0/0
TACE (none/1/2/3/4/≥5 [times])	8/11/9/7/8/18	23/14/9/2/13	48/6/3/0/1/3
Radiation therapy (none/1/≥2 [times])	45/13/3	53/6/2	51/9/1
Other molecular target drugs (no/yes)	61/0	61/0	56/5 *

* Lenvatinib was used for 4 cases and regorafenib for 1 case. RFA, radiofrequency ablation; TACE, transcatheter arterial chemoembolization.

**Table 3 cancers-12-01795-t003:** Univariate and multivariate analyses of possible prognostic factors in the patients, according to the Cox proportional-hazards model.

Variables	Univariate Analysis		Multivariate Analysis	
	HR (95% CI)	*p*-Value	HR (95% CI)	*p*-Value
Sex (male vs. female)	0.872 (0.365–2.082)	0.758		
Age (years)	0.980 (0.956–1.006)	0.127		
Etiology (HCV vs. HBV)	0.519 (0.245–1.101)	0.088		
Etiology (others vs. HBV)	0.904 (0.423–1.931)	0.795		
Child–Pugh score (6 vs. 5)	1.630 (0.850–3.154)	0.141		
Bone metastasis (yes vs. no)	1.077 (0.475–2.445)	0.859		
Cancer stage (III vs. IV)	1.009 (0.526–1.935)	0.979		
Combination therapy (yes vs. no)	1.007 (0.508–1.995)	0.984		
Therapeutic effect (PD vs. CR/PR/SD)	3.044 (1.624–5.705)	<0.001	3.633 (1.811–7.286)	<0.001
BMI (kg/m^2^)	0.954 (0.846–1.075)	0.440		
Sarcopenia (yes vs. no)	1.903 (1.057–3.428)	0.032	1.330 (0.653–2.709)	0.432
∆SMI_pre_ (RD vs. non-RD groups)	0.835 (0.394–1.770)	0.639		
∆SATI_pre_ (RD vs. non-RD groups)	0.797 (0.390–1.628)	0.533		
∆VATI_pre_ (RD vs. non-RD groups)	1.001 (0.490–2.046)	0.998		
∆SMI_during_ (RD vs. non-RD groups)	3.190 (1.460–6.967)	0.004	1.515 (0.609–3.772)	0.372
∆SATI_during_ (RD vs. non-RD groups)	3.061 (1.530–6.122)	0.002	3.366 (1.362–8.321)	0.009
∆VATI_during_ (RD vs. non-RD groups)	2.455 (1.241–4.857)	0.010	1.477 (0.786–2.774)	0.226
∆SMI_post_ (RD vs. non-RD groups)	2.239 (0.798–6.286)	0.126		
∆SATI_post_ (RD vs. non-RD groups)	2.239 (0.798–6.286)	0.126		
∆VATI_post_ (RD vs. non-RD groups)	1.280 (0.463–3.535)	0.634		

HR, hazard ratio; HBV, hepatitis B virus; HCV, hepatitis C virus; CI, confidence interval; PD, progressive disease; CR, complete response; PR, partial response; SD, stable disease; BMI, body mass index; RD, rapid depletion; MD, mild depletion; SMI, skeletal muscle index; SATI, subcutaneous adipose tissue index; VATI, visceral adipose tissue index.

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
