# Peer review of "Rapid Depletion of Subcutaneous Adipose Tissue during Sorafenib Treatment Predicts Poor Survival in Patients with Hepatocellular Carcinoma"

_cancers, 2020, doi:10.3390/cancers12071795_

Round 1
Reviewer 1 Report
The manuscript of Imai et al. brings information regarding subcutaneous adipose tissue lost in patients with hepatocellular carcinoma receiving sorafenib. Computed tomography (CT) analysis were performed to evaluate tissue lost over treatment time. I have made some comments and suggestions to help improve the manuscript. However, it does not adequately discuss the available information in the field of body composition.
Major concerns:
The CT analysis allows to evaluated adipose tissue compartments not fat, so the word fat should not be used. Instead, adipose tissue (AT) is more appropriated.
Presence of sarcopenia must be considered as the obesity paradox does not seem to be applied for patients with low skeletal muscle index (SMI) and obesity. Furthermore, low SMI is a predictor of poor outcome in patients with cancer.
Cancer-associated cachexia was not considered as one of the causes of subcutaneous AT and skeletal muscle wasting. Why weight loss was not reported?
An adiposity index has been previously described and was not considered in the manuscript. I would suggest to classify patients by these sex-specific cutoffs
AT body distribution differs by sex (doi: 10.1210/edrv.21.6.0415) and this should be considered in the analysis using sex-specific cutoffs.
Therefore, I would like to suggest to include initial stage SMI and adiposity index divided by sex in table 1. Furthermore, they could be characterized by low or normal following stablished cutoffs (SMI: doi: 10.1200/JCO.2012.45.2722 or http://dx.doi.org/10.1016/j.clnu.2016.07.015 and subcutaneous and visceral AT index: http://dx.doi.org/10.1016/j.clnu.2016.07.015)
Why muscle radiodensity were not presented?
Methods: CTs analysis must be better described, I checked the referenced article [32] and no distinction of visceral and subcutaneous adipose tissue was described there.
The therapeutic effects categories could be described in the Methods.
Minor comments:
Title of the manuscript could be more informative and less vague.
Table 1: as a suggestion, abbreviations could be in the end of the table, not in the title.
Table 1: “Median survival days after introducing sorafenib (months)”: days or months?
Line 229 – Figure instead of “tabure”.
Author Response
Response to Reviewer #1
We are pleased that in the overall comments this reviewer found our study is of interest. We also thank this reviewer’s constructive comments which were most helpful to improve our manuscript. We accordingly revised the manuscript as follows.
Major concerns:
1. The CT analysis allows to evaluated adipose tissue compartments not fat, so the word fat should not be used. Instead, adipose tissue (AT) is more appropriated.
All of the subcutaneous fat mass (SFM) and visceral fat mass (VFM) in this manuscript are replaced to subcutaneous adipose tissue (SAT) and visceral adipose tissue (VAT), respectively. Thank you for your kind suggestion.
2. Presence of sarcopenia must be considered as the obesity paradox does not seem to be applied for patients with low skeletal muscle index (SMI) and obesity. Furthermore, low SMI is a predictor of poor outcome in patients with cancer.
According to the Japan Society of Hepatology (JSH) guidelines for sarcopenia (new reference # 32), we define sarcopenia as SMI value of ≤38.0 cm2/m2 for women and ≤42.0 cm2/m2 for men (lines 83-84, and 232-233). We add the presence of sarcopenia to new Table 1 and 3. The prevalence of sarcopenia was 41% (25/61) (Table 1), and the presence of sarcopenia affected survival by univariate analysis although it did not by multivariate analysis (Table 3). We also make reference to sarcopenia in the discussion section (lines 186-187). We thank your valuable suggestion.
3. Cancer-associated cachexia was not considered as one of the causes of subcutaneous AT and skeletal muscle wasting. Why weight loss was not reported?
As you say, body weight is the most convenient and non-invasive indicator of nutritional assessment. Loss of body weight is also closely associated with cancer-associated cachexia. Therefore, the data of body composition obtained by CT images should be compared with the changes of body weight in future study. We pointed out the lack of body weight assessment as one of the limitations in our study (lines 194-198). Nevertheless, we believe that the assessments of body composition using sequential CT images in this study are extremely useful because it can avoid the effect of gained weight related to edema, ascites, and pleural effusion which are often seen in patients with advanced HCC. We specified this advantage in the Discussion section (lines 151-153).
4. An adiposity index has been previously described and was not considered in the manuscript. I would suggest to classify patients by these sex-specific cutoffs
AT body distribution differs by sex (doi: 10.1210/edrv.21.6.0415) and this should be considered in the analysis using sex-specific cutoffs.
Following this suggestion, the enrolled patients were divided into two groups based on the 20th percentile cutoffs for each gender. The patients within the 20th percentile cutoffs were classified into a rapid depletion (RD) group and those who did not meet the criteria were classified into a non-rapid depletion (non-RD) group, citing new reference #20 (lines 118-123). We appreciate your valuable comment again.
5. Therefore, I would like to suggest to include initial stage SMI and adiposity index divided by sex in table 1. Furthermore, they could be characterized by low or normal following stablished cutoffs (SMI: doi: 10.1200/JCO.2012.45.2722 or http://dx.doi.org/10.1016/j.clnu.2016.07.015 and subcutaneous and visceral AT index: http://dx.doi.org/10.1016/j.clnu.2016.07.015)
According to this suggestion, the variables in Table 1 were divided by sex. Female patients were significantly younger (p= 0.022), had more SATI (p < 0.001), and less hepatitis C virus (HCV)-related HCC (p = 0.042) when compared male patients (lines 79-80). We used JSH guidelines for sarcopenia (reference #35) because all the patients are Japanese, and annualized changes in body composition were divided into rapid and non-rapid group based on the 20th percentile cutoffs for each gender described above. New Table 3 demonstrated that therapeutic effect and rapid depletion of SATI during sorafenib treatment were independent predictors of survival in HCC patients treated with sorafenib (lines 123-129). The Kaplan-Meier methods revealed that patients with RD group in ΔSATIduring (<-32.51 for women and <-15.49 for men) experienced a significantly poorer survival according to the log-lank test. (lines 137-138, Figure 2b). Thanks to your valuable suggestion, the rapid depletion of skeletal muscle, SAT, and VAT were more clearly defined, and our conclusions were also more trustworthy.
6. Why muscle radiodensity were not presented?
As you say, low muscle attenuation was reported to be a prognostic factor of patients with HCC. We pointed out this as one of the limitations in our study, citing new reference #33 (lines 198-202).
7. Methods: CTs analysis must be better described, I checked the referenced article [32] and no distinction of visceral and subcutaneous adipose tissue was described there.
A built-in function in the SYNAPSE VINCENT software can automatically detect CT level at the umbilical point, and also automatically measure the cross-sectional area of SAT and VAT at that CT level. We added this description in the Materials and Methods section (lines 235-237).
8. The therapeutic effects categories could be described in the Methods.
According to the Response Evaluation Criteria in Cancer of the Liver, CR, PR, SD, and PD are defined as a 100% tumor-necrotizing effect or a complete reduction in tumor size, a tumor-necrotizing effect or tumor size reduction rate between 50% and <100%, effects other than PR and PD, and tumor growth >25% regardless of the necrotizing effect or emergence of a new lesion, respectively. We added this description in the Materials and Methods section (lines 215-218).
Minor comments:
1. Title of the manuscript could be more informative and less vague.
We changed the title of the manuscript to “Rapid depletion of subcutaneous adipose tissue during sorafenib treatment predicts poor survival in patients with hepatocellular carcinoma” (lines 2-4).
2. Table 1: as a suggestion, abbreviations could be in the end of the table, not in the title.
We modified the titles of tables and figures according to this suggestion. We thank your kind suggestion.
3. Table 1: “Median survival days after introducing sorafenib (months)”: days or months?
We changed “days” to “time” in Table 1
4. Line 229 – Figure instead of “tabure”.
We checked all the typographical errors in this manuscript, and got English proofreading for the revised version.

Reviewer 2 Report
This study illustrates changes in body composition in patients with HCC treated with sorafenib using CT imaging. Tables and figures are simple so that results are easy to understand. There are several points to be poited, some of which may be fundamental.
Abstract:
The second last line:
What does "rapid depletion" mean? And how "often " was it seen? There are no statement elsewhere in the abstract regarding how is rapid depletion and how is slow depletion. There may not be anything regardind the frequency in SFM depletion, either. These statements are not scientific.
Introduction:
lines 40-41:
The author says "weight loss" "can easily cause malnutrition". What does it mean? I cannot understand why weight loss cause malnutrition. I think malnutrition may cause weight loss, instead.
Table 1:
Why etiology (HBV, HCV, and others) are not included in table 3?
Table 2:
Is there any other chemotherapy than sorafenib?
Table 3:
The most important limitation of this study is lack of data regarding changes in body weight. The author proposed SFM depletion as a prognostic factor. However, when SFM is reduced, body weight is also reduced, of course. In addition, body weight is much more easily available than SFM measured by CT scan. So if there is no additional usufulness in SFM measurement to weight change, there is no reason to measure SFM repeatedly in multiple CT scan.
Discussion:
lines 150-151: Why the author think alleviation of depletion is "useful to improve survival" and understanding of the chages in body composition is "very important" only from the results of observational study? As many as 7 references [13-17, 20, 21], all of them are observational study.
If the authors have already known that "understanding of the chages in body composition is very important", why did they conduct this study?
lines 174-183: The author says both of malnutrition and overnutrition are harmful. How do they determine "adequate" nutritional state? Statement regarding malnutrition, overnutrition, and adequate nutrition without refering definition does not make sense.
lines187-189: Why is it unclear whether the cutoff valures are optimal?
Material and Methods:
It is nuclear how the interval of sorafenib treatment was determined.
Conclusions:
line 242: What does "unfavorable changes in body composition" mean?
lines 243-246: The statement "we should not overlook this sign and should take proper measure" and "it might be important to try to maintain adequate body compositions" are not derived from the results of the present study and should not be included in "conclusion" of the study. In addition, on lines 199-201 (Material and methods) the author says "when appetite loss or signs of malnutrition associatedd with sorafenib treatement wer observed, nutrition therapy and administration of antiemetic were starded as needed". It means that poor prognosis in accerelated SFM depletion was a result of the nutritional therapy, which denotes "to try to maintain adequate body compositions" may not "be important".
Author Response
Response to Reviewer #2
We are pleased that in the overall comments this reviewer found our study is of interest. We also thank this reviewer’s constructive comments which were most helpful to improve our manuscript. We accordingly revised the manuscript as follows.
Abstract:
1. The second last line:
What does "rapid depletion" mean? And how "often " was it seen? There are no statement elsewhere in the abstract regarding how is rapid depletion and how is slow depletion. There may not be anything regardind the frequency in SFM depletion, either. These statements are not scientific.
The enrolled patients were divided into two groups based on the 20th percentile cutoffs for each gender. The patients within the 20th percentile cutoffs were classified into a rapid depletion (RD) group and those who did not meet the criteria were classified into a non-rapid depletion (non-RD) group (lines 20-22, and 119-123). We also revised the abstract so that vague expressions such as “often” were avoided (lines 27-28). Thanks to your valuable suggestion, the rapid depletion of skeletal muscle, SAT, and VAT were more clearly defined, and our conclusions were also more trustworthy.
Introduction:
2. lines 40-41:
The author says "weight loss" "can easily cause malnutrition". What does it mean? I cannot understand why weight loss cause malnutrition. I think malnutrition may cause weight loss, instead.
As you say, this expression was not correct, so we modified the description (lines 41-43). Your kind indication made the readers to understand our manuscript more easily.
Table 1:
3. Why etiology (HBV, HCV, and others) are not included in table 3?
Following this suggestion, we included etiology in Table 3. It demonstrated that etiology did not affect survival (Table 3).
Table 2:
4. Is there any other chemotherapy than sorafenib?
Other molecular target drugs in Table 2 meant lenvatinib for 4 cases and regorafenib for 1 case. We added this information to the end of Table 2 (lines 110)
Table 3:
5. The most important limitation of this study is lack of data regarding changes in body weight. The author proposed SFM depletion as a prognostic factor. However, when SFM is reduced, body weight is also reduced, of course. In addition, body weight is much more easily available than SFM measured by CT scan. So if there is no additional usufulness in SFM measurement to weight change, there is no reason to measure SFM repeatedly in multiple CT scan.
As you say, body weight is the most convenient and non-invasive indicator of nutritional assessment. Loss of body weight is also closely associated with cancer-associated cachexia. Therefore, the data of body composition obtained by CT images should be compared with the changes of body weight in future study. We pointed out the lack of body weight assessment as one of the limitations in our study (lines 194-198). Nevertheless, we believe that the assessments of body composition using sequential CT images in this study are extremely useful because it can avoid the effect of gained weight related to edema, ascites, and pleural effusion which are often seen in patients with advanced HCC. We emphasized this point in the Discussion section (lines 151-153).
Discussion:
6. lines 150-151: Why the author think alleviation of depletion is "useful to improve survival" and understanding of the changes in body composition is "very important" only from the results of observational study? As many as 7 references [13-17, 20, 21], all of them are observational study.
If the authors have already known the "understanding of the changes in body composition is very important", why did they conduct this study?
In all of the previous studies, evaluation of body composition using CT images was performed only once or twice. Therefore, this is the first study using sequential CT examinations to elucidate how body composition changes from the initial stage to the terminal stage of HCC patients treated with sorafenib, and to assess the association between these changes and survival. Furthermore, Sequential CT examinations revealed that skeletal muscle mass, SAT, and VAT decreased over time in HCC patients, annualized depletion rates of SMI and VATI were greatest after withdrawal of sorafenib, but depletion of SATI had already accelerated during sorafenib treatment, and that rapid depletion of SATI during sorafenib treatment was an independent predictor together with therapeutic effect. We emphasized this advantage over the previous studies in the Discussion section (lines 147-151), and specified these results in the Conclusions section (lines 260-264).
7. lines 174-183: The author says both of malnutrition and overnutrition are harmful. How do they determine "adequate" nutritional state? Statement regarding malnutrition, overnutrition, and adequate nutrition without refering definition does not make sense.
Rapid depletion of SAT (∆SATI: < -32.51 for women and < -15.49 for men) during sorafenib treatment implies malnutrition, and these cutoff values can be used as a biomarker of malnutrition associated with poor survival in HCC patients. Our previous study (reference #31) revealed that patients with higher than 47.2 cm2/m2 of VATI had significantly higher risk of recurrence of HCC after curative treatment. Sarcopenia, which was defined as an SMI value ≤ 38.0 cm2/m2for women and ≤ 42.0 cm2/m2 for men in Japan, is also a significant prognostic factor of HCC. We suggested that these cutoff values of body composition may be used as indicators of appropriate nutritional therapy in the Discussion section (lines 179-190). Thanks to your valuable indication, we think that our suggestion concerning appropriate nutritional therapy got substantially a definite form, and also useful in a clinical situation.
8. lines187-189: Why is it unclear whether the cutoff valures are optimal?
We newly defined patients within 20th percentile of annualized changes in body composition for each gender as a rapid group (lines 118-123), and rapid depletion of SATI during sorafenib treatment (∆SATI: < -32.51 for women and < -15.49 for men) became a much more suitable prognostic factor (Table 3 and Figure 3b). Thus, we delated all this description from the limitations.
Material and Methods:
9. It is nuclear how the interval of sorafenib treatment was determined.
We had already described that it was determined “from the introduction of sorafenib to the cancellation of the agent” (lines 240-241).
Conclusions:
10. line 242: What does "unfavorable changes in body composition" mean?
From this study and previous studies, we think that the rapid depletion of SATI during sorafenib treatment (∆SATI: < -32.51 for women and < -15.49 for men), the excess accumulation of VAT (VATI >47.2), and the presence of sarcopenia (SMI ≤ 38.0 for women and ≤ 42.0 for men in Japan), may imply “unfavorable changes in body composition”. We referred to these cutoff values associated with poor survival in HCC patients in the last of the Discussion section (lines 179-190).
11. lines 243-246: The statement "we should not overlook this sign and should take proper measure" and "it might be important to try to maintain adequate body compositions" are not derived from the results of the present study and should not be included in "conclusion" of the study. In addition, on lines 199-201 (Material and methods) the author says "when appetite loss or signs of malnutrition associated with sorafenib treatment were observed, nutrition therapy and administration of antiemetic were started as needed". It means that poor prognosis in accerelated SFM depletion was a result of the nutritional therapy, which denotes "to try to maintain adequate body compositions" may not "be important".
As you say, we did not take adequate nutritional therapy to our patients in this study because accelerated SAT depletion was seen during sorafenib treatment. Thus, we moved this description to the Discussion section (lines 173-175). Furthermore, we rewrote the conclusions so that only the findings derived from the results of this study would be written (lines 260-264).
Thanks to all of your valuable comments, vague expressions can be avoided throughout our study, “rapid depletion” can be clearly defined. Furthermore, the result we want to demonstrate the most in this study can become more clearer as follows; rapid depletion of subcutaneous adipose tissue during sorafenib treatment predicts poor survival in patients with hepatocellular carcinoma. We appreciate your valuable comments again and again.

Reviewer 3 Report
The article is well written and authors explain in a retrospective study the changes in body composition in HCC patients treated with sorafenib. The results and the discussion chapters are complete and with correct in their reasoning.
In my opinion there were only few minor changes that authors could apport to complete their work.
First, only minor spell check of language are required.
Second, in the Introduction chapter, a minimal reference to the contribution of the presence of bone lesions to changes in body composition could be inserted. The presence of bone lesions in HCC results in poorer disease control, also in patients treated with sorafenib and opioids. Several articles showed that bone lesions can precipitate disease evolution and liver and skeletal muscles composition/activity. Are there patients with HCC and bone lesions in the 61 patients tested in their study? If yes, authors could evaluate the possibility to compare changes in body composition between HCC patients with/without bone lesions.
Best regards
Author Response
Response to Reviewer #3
We are pleased that in the overall comments this reviewer found our study is of interest. We thank this reviewer’s constructive comments which strengthen the conclusion of our study.
1. First, only minor spell check of language are required.
We checked all the typographical errors in this manuscript, and got English proofreading for the revised version.
2. Second, in the Introduction chapter, a minimal reference to the contribution of the presence of bone lesions to changes in body composition could be inserted. The presence of bone lesions in HCC results in poorer disease control, also in patients treated with sorafenib and opioids. Several articles showed that bone lesions can precipitate disease evolution and liver and skeletal muscles composition/activity. Are there patients with HCC and bone lesions in the 61 patients tested in their study? If yes, authors could evaluate the possibility to compare changes in body composition between HCC patients with/without bone lesions.
We had nine patients with bone lesions, and the presence of bone lesions did not affect survival in this study. We referred to these results in the Introduction section (lines 51-52), Table 1 and 3, citing new reference #13 that suggested the association between bone lesions and poor survival. We appreciate your valuable comment.

Round 2
Reviewer 1 Report
The manuscript was significantly improved after the first review.
I have some minor comments:
Study limitation should also include the lack of muscle function assessment (as hand grip strength) and the lower number of female patients enrolled.
Duration of sorafenib treatment is still unclear.
Figure 1 could have a better visual aspect and standard error could be used instead of standard deviation (in fact, this information should be in the figure legend).
Figure 2 – missing standard error bars. Legends should be more informative regarding statistical analysis.
Line 45: “contributing to tissue wasting” or “cancer-associated cachexia”, only “wasting” is not appropriated.
Line 150 – “assess the association” could be improved
Lines 172-174 – please check “If appetite loss or signs of malnutrition are associated with sorafenib treatment were observed, appropriate nutrition therapy and the administration of antiemetics should be started as neededused”.
Author Response
Response to Reviewer #1
We are pleased that this reviewer found our revised manuscript was significantly improved. We also thank this reviewer’s constructive comments which were most helpful to improve our manuscript. We accordingly revised the manuscript as follows.
Study limitation should also include the lack of muscle function assessment (as hand grip strength) and the lower number of female patients enrolled.
According to this suggestion, we add these points as study limitations in the Discussion, with citing a new reference # 31 that demonstrated that grip strength is associated with a prognostic factor for patients with liver cirrhosis (lines 192 and 200-203). We thank your valuable suggestion.
Duration of sorafenib treatment is still unclear.
We decided to discontinue this agent if PD was observed for a certain period of time or severe adverse events occurred. In fact, sorafenib was discontinued because of PD (n=22); severe adverse events, including HFS (n=4), general fatigue (n=4), ascites (n=4), liver dysfunction (n=4), and appetite loss (n=4); and other reasons. The median duration of sorafenib treatment was 15.7 months (95% confidence interval [CI] 12.3–19.0 months). We added these information as to duration of sorafenib treatment in the Results (lines 76-79), Materials and Methods (lines 221-223) section, and Table 1.
Figure 1 could have a better visual aspect and standard error could be used instead of standard deviation (in fact, this information should be in the figure legend).
Figure 2 – missing standard error bars. Legends should be more informative regarding statistical analysis.
According to this suggestion, standard errors were added in Figure 1 and 2, and we clarified that error bars represent standard errors in the figure legend.
Line 45: “contributing to tissue wasting” or “cancer-associated cachexia”, only “wasting” is not appropriated.
We changed the word of “wasting” to “cancer-associated cachexia” (lines 45). We thank your kind indication.
Line 150 – “assess the association” could be improved
According to this suggestion, we modified the relevant part of a writing so that the readers can understand more easily (lines 153-154).
Lines 172-174 – please check “If appetite loss or signs of malnutrition are associated with sorafenib treatment were observed, appropriate nutrition therapy and the administration of antiemetics should be started as neededused”.
We corrected typographical errors in the relevant part (lines 177-178). Thank you for your kind indication.

Reviewer 2 Report
The manuscript is generally well-corrected.
There are several comments to be addressed.
#1
Discussion: lines 179-190
The author stated that low SAT, high VAT, and low SMI are associated with poor outcome. However, these are the results from the observational studies. The author still misunderstand the results from the observational study. The results from the observational study does not always imply causality. The statement " necessary to perform appropriate nutritional therapy ..." is not always correct. The easy explanation is that the "nutritional therapy" may not have only the effect to raise or reduce such body mass but have any other effect, some of which may have adverse effect. The same misunderstanding is seen in lines 178-179.
#2
Material and Methods: (the same part as the comment #9 of reviewer #2)
When the reviewer asked "it is nuclear how the interval of sorafenib treatment was determined", the authors replied, "we had already described that it was determined “from the introduction of sorafenib to the cancellation of the agent” (lines 240-241).". This is not what is asked. It is still uncelar how the timing of the cancellation of the agent was determined. The question is, some factors which force the doctors to extend or shorten the treatment period may affect the outcome.
Author Response
Response to Reviewer #2
We are pleased that this reviewer found our revised manuscript was generally well-corrected. We also thank this reviewer’s constructive comments which were most helpful to improve our manuscript. We accordingly revised the manuscript as follows.
#1
Discussion: lines 179-190
The author stated that low SAT, high VAT, and low SMI are associated with poor outcome. However, these are the results from the observational studies. The author still misunderstand the results from the observational study. The results from the observational study does not always imply causality. The statement " necessary to perform appropriate nutritional therapy ..." is not always correct. The easy explanation is that the "nutritional therapy" may not have only the effect to raise or reduce such body mass but have any other effect, some of which may have adverse effect. The same misunderstanding is seen in lines 178-179.
As you say, the relevant part may cause misunderstanding. We deleted ambiguous expressions to cause misunderstanding, and toned down the whole expressions of the relevant part (lines 182-190). We appreciate your valuable comment.
#2
Material and Methods: (the same part as the comment #9 of reviewer #2)
When the reviewer asked "it is nuclear how the interval of sorafenib treatment was determined", the authors replied, "we had already described that it was determined “from the introduction of sorafenib to the cancellation of the agent” (lines 240-241).". This is not what is asked. It is still uncelar how the timing of the cancellation of the agent was determined. The question is, some factors which force the doctors to extend or shorten the treatment period may affect the outcome.
We decided to discontinue this agent if PD was observed for a certain period of time or severe adverse events occurred. In fact, sorafenib was discontinued because of PD (n=22); severe adverse events, including HFS (n=4), general fatigue (n=4), ascites (n=4), liver dysfunction (n=4), and appetite loss (n=4); and other reasons. The median duration of sorafenib treatment was 15.7 months (95% confidence interval [CI] 12.3–19.0 months). We added these information as to duration of sorafenib treatment in the Results (lines 76-79), Materials and Methods (lines 221-223) section, and Table 1.
